# Multi-Scale Feature Attention Network for Rapid and Non-Destructive Quantification of Aflatoxin B_1_ in Maize Using Hyperspectral Imaging

**DOI:** 10.3390/foods14213769

**Published:** 2025-11-03

**Authors:** Yichi Zhang, Kewei Huan, Xiaoxi Liu, Yuqing Fan, Xianwen Cao, Xueyan Han

**Affiliations:** 1College of Physics, Changchun University of Science and Technology, Changchun 130022, China; yc15147141357@163.com (Y.Z.); fanyuqing@cust.edu.cn (Y.F.); 2023800011@cust.edu.cn (X.C.); xueyanbest@163.com (X.H.); 2Institute of Scientific and Technical Information of Jilin, Changchun 130033, China; liuxiaoxiyouxiang@126.com

**Keywords:** near-infrared spectroscopy, hyperspectral technology, aflatoxin B_1_, maize, quantitative analysis

## Abstract

Maize, a globally important crop, is highly susceptible to aflatoxin contamination, posing a serious threat. Therefore, accurate detection of aflatoxin levels in maize is of critical importance. In this study, the Multi-Scale Feature Network with Efficient Channel Attention (MSFNet-ECA) model, based on near-infrared hyperspectral imaging combined with deep learning techniques was developed to analyze the content of aflatoxin B_1_ (AFB_1_) in maize. Three data augmentation methods—multiplicative random scaling, bootstrap resampling, and Wasserstein generative adversarial networks (WGAN)—were compared with various preprocessing strategies to assess their impact on model performance. Multiplicative random scaling combined with second derivative (D2) preprocessing yielded the best predictive performance for the MSFNet-ECA model. Using this augmentation, the MSFNet-ECA model outperformed four conventional models (partial least squares regression (PLSR), support vector regression (SVR), extreme learning machine (ELM), and one-dimensional convolutional neural network (1D-CNN)), achieving a root mean square error of prediction (RMSEP) of 2.3 μg·kg^−1^, coefficient of determination for prediction ($R_{p}^{2}$) of 0.99, and the residual predictive deviation (RPD) of 9, with accuracy improvements of 86.4%, 79.1%, 71.3%, and 42.5%, respectively. This finding demonstrates that applying data augmentation methods substantially improves the predictive performance of hyperspectral chemometric models driven by deep learning. Moreover, when combined with data augmentation techniques, the proposed MSFNet-ECA model can accurately predict AFB_1_ content in maize, offering an efficient and reliable tool for hyperspectral applications in food quality and safety monitoring.

## 1. Introduction

As a crop of high nutritional and economic relevance, maize is cultivated on a global scale. It provides an essential food source for humans and livestock and is also used to produce bioethanol, edible oils, starch, and other derivatives [1]. Nevertheless, maize is highly susceptible to fungal infections by *Fusarium* and *Aspergillus* species during storage. Among these, aflatoxin B_1_ (AFB_1_) is a potent and carcinogenic metabolite generated by *Aspergillus flavus* and *Aspergillus parasiticus*, which represents a serious health hazard. Long-term low-dose exposure to AFB_1_ may cause gradual immunosuppression and significantly elevate cancer risk [2,3]. According to the International Agency for Research on Cancer (IARC), AFB_1_ is categorized as a Group 1 carcinogen affecting humans and animals [4]. Consequently, many countries have established strict regulations on the permissible levels of AFB_1_ in maize. The U.S. Food and Drug Administration (FDA) stipulates that the AFB_1_ concentrations in maize and maize products for human consumption must not exceed 20 μg·kg^−1^ [5]. In China, the National Standard GB 2761-2017 requires that AFB_1_ levels in maize-based foods remain under 20 μg·kg^−1^. Therefore, monitoring the AFB_1_ concentrations in maize is of critical importance to prevent health hazards to humans and animals caused by mold-contaminated maize.

Conventional methods for detecting AFB_1_ in crops include high-performance liquid chromatography, thin-layer chromatography, and enzyme-linked immunosorbent assay [6]. These methods are characterized by high accuracy, good selectivity, and low detection limits. However, they also present several drawbacks, such as complex operation procedures, long detection times, and destructive effects on samples, which restrict their use in quick and non-invasive detection tasks. Consequently, ensuring real-time monitoring and sustaining food safety quality control across the stages of crop production, storage, and distribution necessitates the development of highly efficient, rapid, and non-destructive detection strategies [7,8,9].

In recent years, near-infrared (NIR) spectroscopy has gained attention as a rapid and non-destructive analytical method. It requires no elaborate sample preparation. It is broadly applied to the analysis of food and various agricultural commodities [10,11]. The spectral information obtained from NIR technology primarily originates from the overtone and combination band absorptions of molecular vibrations such as C-H, N-H, and O-H bands [12]. However, conventional NIR spectroscopy mainly captures the overall spectral characteristics of a sample and lacks the ability to describe its spatial distribution, thus limiting its accuracy and sensitivity when analyzing heterogeneous or complex samples.

Hyperspectral imaging (HSI) technology integrates imaging and spectroscopy, offering advantages such as a wide spectral range and high spectral resolution. It can simultaneously capture both the internal compositional information and the external morphological features of a sample in a single measurement, enabling precise analysis and characterization of the sample [13,14]. HSI is extensively utilized in various domains, such as food quality assessment, agricultural studies, and environmental monitoring [15,16,17,18]. Prior research indicates that integrating HSI with chemometric modeling enables non-invasive detection of mycotoxins in food, and offers a reliable and effective solution for monitoring grain safety. For example, Yong-Kyoung Kim applied multiple HSI techniques combined with discriminant analysis and support vector machine algorithms to efficiently and accurately detect aflatoxins in maize, achieving a 95.7% classification accuracy for both short-wave infrared and fluorescence models [19]. Gayatri Mishra utilized NIR hyperspectral imaging combined with competitive adaptive reweighted sampling and partial least squares regression (PLSR) to achieve rapid and non-destructive detection of AFB_1_ content in almonds, with the prediction model achieving R^2^ of 0.95 and RMSEP of 0.09 μg·g^−1^ [20]. Despite the comprehensive spectral and spatial data provided by HSI, traditional chemometric methods are limited in their ability to model the complex, nonlinear interactions between NIR spectra and the chemical makeup of samples [21]. Consequently, a key challenge in current research lies in efficiently identifying spectral features strongly correlated with AFB_1_ content and constructing predictive models with high generalization performance.

Recently, deep learning technology has advanced rapidly and been extensively adopted in domains including image processing, speech recognition, and natural language processing [22,23,24]. With its end-to-end feature extraction ability and strong nonlinear modeling capacity, deep learning exhibits significant potential for applications in agricultural and food safety detection [25,26]. To detect AFB_1_ in agricultural products, Hongfei Zhu utilized a one-dimensional residual temporal convolutional network for pixel-level aflatoxin identification in peanuts through hyperspectral imaging. The model achieved training and testing accuracies of 99.60% and 99.26%, respectively [27]. Hui Jiang developed a two-dimensional convolutional neural network that integrates Gramian Angular Summation Field image encoding combined with NIR spectral data to detect AFB_1_ content in moldy peanuts. The resulting GASF-2D-CNN model achieved RMSEP of 2.0 μg·kg^−1^, $R_{p}^{2}$ of 0.99, and RPD of 8.3 [28]. Although deep learning demonstrates outstanding performance in AFB_1_ detection, it heavily relies on high-quality training samples. The complexity and time consuming nature of acquiring hyperspectral data along with corresponding compound concentrations make it challenging to obtain sufficiently large and well distributed spectral datasets. Consequently, effectively leveraging limited experimental data to enhance model generalization remains a major challenge in hyperspectral data modeling.

Data augmentation is a technique for enhancing the robustness and training performance of neural networks by increasing the diversity of training samples through various transformations applied to the original data, without changing their labels [29]. In recent years, data augmentation has been extensively used in NIR spectral analysis as an effective approach to alleviate problems such as small sample sizes and uneven sample distribution [30,31,32]. In this study, data augmentation was applied to expand the maize hyperspectral dataset, enhancing its diversity and representativeness while facilitating stable training of the deep learning model.

Here, we developed a Multi-Scale Feature Network with Efficient Channel Attention (MSFNet-ECA) model built upon deep learning, enhanced by effective data augmentation, for quantitative analysis of AFB_1_ in maize using NIR hyperspectral imaging. Through designed maize mold experiments, we collected NIR hyperspectral images of maize samples with different degrees of mold, and measured their corresponding AFB_1_ content to build a sample database. Data augmentation was further applied to expand the original spectral dataset and enhance its diversity. Experimental results demonstrated that multiplicative random scaling combined with second-derivative (D2) preprocessing yielded the best predictive performance for the MSFNet-ECA model. Using multiplicative random scaling as the data augmentation strategy, we systematically developed four conventional quantitative models—PLSR, support vector regression (SVR), extreme learning machine (ELM), and one-dimensional convolutional neural network (1D-CNN)—and compared their performance with that of the proposed MSFNet-ECA model. Results demonstrated that the MSFNet-ECA model enhanced with multiplicative random scaling achieved optimal predictive accuracy, yielding RMSEP of 2.3 μg·kg^−1^, $R_{p}^{2}$ of 0.99, and RPD of 9 Compared to conventional models, MSFNet-ECA improved modeling accuracy by 86.4% (PLSR), 79.1% (SVR), 71.3% (ELM), and 42.5% (1D CNN), respectively. Experimental results confirm that integrating data augmentation with the MSFNet-ECA model enables accurate prediction of AFB_1_ levels in maize. This approach provides an effective and dependable framework for hyperspectral imaging-based food quality and safety monitoring.

## 2. Materials and Methods

### 2.1. Preparation of Moldy Maize

Yellow waxy maize kernels were purchased online in bulk from Qiqihar City, Heilongjiang Province, China. To ensure the freshness of the samples and avoid unintended fungal contamination prior to the experiment, the maize kernels were obtained in multiple batches, approximately 5 kg per batch. In addition, all purchased samples were preliminarily screened to confirm the absence of AFB_1_ contamination before use. To establish conditions conducive to natural fungal growth and AFB_1_ production, 1 kg of maize kernels was evenly spread on a stainless-steel tray, which served solely as an inert carrier during incubation. The tray was then placed in a controlled-environment incubator (LHS-HC-III, Shanghai Yiheng Scientific Instruments Co., Ltd., Shanghai, China). The temperature and relative humidity were maintained at 30 °C and 90%, respectively. The fungal spores naturally present on the maize kernel surfaces proliferated and produced AFB_1_ under these controlled environmental conditions without the need for artificial inoculation. Each experimental cycle lasted approximately 7–10 days. During incubation, the kernels were manually rotated once per day to ensure homogeneous fungal development across the tray. Multiple samplings were conducted within each experimental cycle, with sampling performed once every two days. At each sampling event, approximately 70 kernels were randomly selected from different positions of the stainless-steel tray to ensure sampling representativeness. After several independent cycles, a total of 330 maize samples with varying AFB_1_ contamination levels were obtained.

### 2.2. Acquisition of Hyperspectral Images of Moldy Maize and Spectral Data Extraction

The hyperspectral images of moldy maize samples in this study were obtained using a push-broom NIR hyperspectral imaging system (Hyperspec III NIR hyperspectral imaging system, Headwall Photonics, Fitchburg, MA, USA). The overall imaging workflow and system configuration are illustrated in Figure 1. The imaging system mainly consists of halogen light sources, a motorized translation stage, a zoom lens, a CCD camera, and a workstation integrated with hyperspectral imaging and data collection software. The system covers a spectral range of 925.4–1701.28 nm with a spectral resolution of 4.76 nm, comprising 164 spectral bands in total. Due to significant noise at both ends of the spectrum, the bands in the range of 1049.16–1653.68 nm were selected for this study, yielding 128 spectral bands in total. Before image acquisition, the camera was calibrated to ensure optimal image quality. Moreover, white and dark reference corrections were applied to the raw images to reduce the impact of sensor dark current and uneven illumination from halogen sources, thereby enhancing spectral fidelity and ensuring robust analytical outcomes. During imaging, the acquisition distance was 20.5 cm, the exposure time was 0.3 ms, the translation stage moved at a speed of 22 mm/s, the halogen light incident angle was 45°, and the frame period was 10 ms. Furthermore, to minimize background interference, the selected maize samples were evenly spread on black cardboard for imaging.

In this study, hyperspectral maize images were processed using ENVI Classic 5.6 (Harris Geospatial Solutions, Inc., Broomfield, CO, USA) to identify and isolate Regions of Interest (ROIs). The selection of the ROI significantly affects spectral feature selection and subsequent analyses. Threshold segmentation is commonly adopted in engineering applications for its straightforward and efficient nature [33]. In this study, the segmentation threshold was calculated using the grayscale histogram. Subsequently, a binarization method was used to separate the maize image from the background, thereby reducing the interference of background pixels on spectral extraction. Finally, the separated maize region was defined as the ROI, and the average spectral data of all pixels within the ROI were extracted to obtain spectral data that can represent the characteristics of the maize samples.

### 2.3. Determination of Aflatoxin B_1_ Content in Moldy Maize

The AFB_1_ concentrations in maize were measured through fluorescence quantitative immunochromatography. The mycotoxin detector (HM-L03, Shandong Hengmei Electronic Technology Co., Ltd., Weifang, Shandong, China) was preheated to 37 °C, while the test card, fluorescent microsphere lyophilized wells, and sample diluent (Shandong Hengmei Electronic Technology Co., Ltd., Weifang, Shandong, China) were equilibrated to room temperature. Following hyperspectral imaging, maize kernels were pulverized and sieved through a 20-mesh screen. Five grams of the resulting powder were measured and transferred into a 50 mL centrifuge tube, then mixed with 40 mL of 50% ethanol–water solution and vortexed at 4000 rpm for 4 min. The suspension was transferred to a 2 mL tube, centrifuged at 4000 rpm for 3 min, and the supernatant was obtained for further analysis. Subsequently, 120 μL of the collected supernatant was combined with 1000 μL of sample diluent, followed by gentle mixing to prepare the test solution. Then, 120 μL of the sample was transferred into the lyophilized microsphere well, mixed thoroughly by repeated pipetting, and 60 μL was immediately dispensed onto the test card. The card was then placed into the mycotoxin detector for measurement.

### 2.4. Data Augmentation and Dataset Division

#### 2.4.1. Data Augmentation

Three data augmentation strategies were applied to expand the original spectral dataset, thereby increasing sample diversity and enhancing model robustness and generalization in the study. For fair comparison, each method increased the sample size from 330 to 1320, and all methods were evaluated under comparable experimental settings.

The multiplicative random scaling method introduces amplitude perturbations by multiplying the original spectrum by a random coefficient ranging from 0.9 to 1.1. This approach preserves the overall spectral shape while simulating physical variations such as instrument baseline drift and environmental fluctuations during measurements [34,35]. Because slight amplitude fluctuations can occur in repeated measurements of the same sample, this method produces data that reflect realistic physical variations. Consequently, it enhances the model’s adaptability and robustness to spectral variability.

The Bootstrap resampling method performs sampling with replacement on the original dataset to create new sample sets, effectively expanding the sample space and capturing internal data variability [36]. Such resampling-based approaches are widely recognized as powerful tools for managing multivariate errors and quantifying uncertainty in spectroscopic data modelling, thereby improving the reliability and interpretability of chemometric models [37]. When the sample size is insufficient for reliable modeling, the Bootstrap method can enhance the model’s generalization ability without introducing external information.

The Wasserstein Generative Adversarial Network (WGAN) trains a generator to learn the latent distribution of real spectral data, thereby generating synthetic spectra with realistic statistical properties. Such a method enhances both the diversity and representational capability of the calibration dataset. Unlike traditional GANs, WGAN leverages the Wasserstein distance to evaluate discrepancies between real and generated data, which leads to smoother gradient updates and ultimately improves sample quality and model stability [32].

#### 2.4.2. Dataset Division

To construct a reliable predictive framework, the SPXY algorithm was used to allocate 70% of the dataset for calibration, with the remaining 30% reserved for model validation and performance assessment. The SPXY algorithm is a sample set partitioning method based on joint distances, which evaluates the similarity between samples by simultaneously considering the Euclidean distances of spectral data (X variables) and chemical values (Y variables) [38]. Compared with the KS algorithm, which only partitions samples based on spectral information, SPXY can better capture the overall sample distribution and ensure that representative samples are included across different concentration or response ranges. This effectively covers the multidimensional feature space and further increases the diversity and sample representativeness, thereby enhancing model stability and reinforcing its generalization capacity.

### 2.5. Spectral Preprocessing

Spectral data are often affected by light scattering, baseline drift, stochastic noise, and various uncontrollable external factors, which can distort the true spectral signal and compromise the predictive performance of the model [20]. To enhance the precision and robustness of subsequent regression modeling and to mitigating nonlinear effects, scattering effects, and noise, this study applied five commonly used preprocessing methods to the spectral data: first derivative (D1), second derivative (D2), standard normal variate (SNV), multiplicative scatter correction (MSC), and Savitzky-Golay (S-G) smoothing. In addition, standardization was performed to ensure that all spectral bands had the same mean and variance. Specifically, D1 can effectively correct baseline drift in the spectra; D2 can simultaneously remove baseline and linear trends, thereby reducing additive and multiplicative effects [39]; SNV performs mean centering and divides by the standard deviation, which can significantly correct light scattering effects and reduce systematic differences between samples; MSC performs linear correction for scattering and absorption differences using a reference spectrum, thereby enhancing spectral comparability and improving model robustness [40]; S-G smoothing can effectively remove spectral edge noise, improve the signal-to-noise ratio, and enhance the reliability of central wavelengths while preserving peak features [21].

### 2.6. Regression Modeling Methods

To evaluate the performance of different regression models in predicting AFB_1_ content in maize, we conducted a comparative analysis using PLSR, SVR, ELM, 1D-CNN, and the MSFNet-ECA model developed in this study.

PLSR projects predictor and response variables into a new latent variable space for accurate regression and has demonstrated strong modeling capability for high-dimensional spectral data with severe multicollinearity [41,42,43]. In this study, the optimal number of latent variables was determined via exhaustive search. SVR exhibits strong generalization ability and robust performance in nonlinear regression tasks, particularly when the sample size is limited [5]. The SVR hyperparameters were optimized using a grid search strategy. ELM offers fast learning speed and strong generalization ability and has been widely applied to hyperspectral data regression and classification due to its simple network structure and low parameterization requirements [44,45]. The optimal number of hidden neurons was determined via exhaustive search, and 150 neurons were selected in this study. In addition, 1D-CNN was constructed, consisting of two convolution–pooling blocks for local spectral pattern extraction, followed by two fully connected layers for regression output.

### 2.7. MSFNet-ECA Model

In this study, the deep learning-based MSFNet-ECA model was constructed, with its architecture illustrated in Figure 2. The network consists of three parallel multi-scale convolution branches (branch3, branch5, branch7), three Efficient Channel Attention (ECA) blocks, a Feature Fusion Module, three one-dimensional convolutional layers (conv1, conv2, conv3), a Flatten layer, and two fully connected regression layers (L1 and L2). The multi-scale branch module comprises one-dimensional convolutional layers with receptive fields of 3, 5, and 7 to extract local features at different scales from one-dimensional spectral sequences. After convolution, each branch is connected to an ECA block, as shown in Figure 3a. By leveraging an adaptive local cross-channel interaction mechanism, the ECA block strengthens feature representations across the channel dimension, enabling the network to better capture critical information. Specifically, the ECA module initially performs global average pooling on the convolutional feature maps to produce a channel descriptor vector. Then, a one-dimensional convolution replaces the conventional fully connected layer to achieve localized cross-channel information interaction, which effectively captures inter-channel dependencies while significantly reducing the number of model parameters. Finally, a Sigmoid activation function is applied to produce channel-wise attention weights, which are used to reweight the input features channel by channel, thereby strengthening the selective expression of salient channel features.

The features from the three branches are merged along the channel axis and subsequently input into the fusion module, as illustrated in Figure 3b. This module is implemented as a multilayer perceptron consisting of two fully connected layers. It first applies global average pooling to extract the global channel representation of the concatenated features. Subsequently, two layers of linear transformation with ReLU activation are employed to achieve nonlinear modeling and weight learning of the importance of features from each branch. Finally, a Softmax function outputs a weight vector, which is used to adaptively fuse multi-scale features. The fused features are then input into three sequential one-dimensional convolutional layers to extract deeper-level representations. These convolutional layers are combined with max-pooling operations to reduce the feature sequence length while retaining key information. After passing through the flatten layer, the feature vector is fed sequentially into two fully connected layers for regression prediction, where L1 and L2 are linear output layers.

During model training, the Adam optimization algorithm was employed for parameter updates, with the initial learning rate configured to 0.001. The mean squared error function was adopted as the loss metric. All convolutional layers utilized the Tanh activation function. The training process was conducted for a maximum of 150 epochs with a batch size of 8. A learning rate decay strategy was applied, halving the rate when no improvement in validation loss was detected across 30 successive epochs. Furthermore, an early stopping mechanism was introduced to terminate training if validation performance remained stagnant for 50 epochs, thereby mitigating the risk of overfitting. A comprehensive summary of the model’s architecture configuration and key hyperparameters is provided in Table 1.

### 2.8. Model Performance Evaluation Metrics

In this study, the model performance was evaluated using the root mean square error of calibration (RMSEC) and prediction (RMSEP), the coefficient of determination for calibration ($R_{c}^{2}$) and prediction ($R_{p}^{2}$), as well as the residual predictive deviation (RPD). The calculation formulas for these metrics are shown in Equations (1)–(3).(1)$$RMSEC,RMSEP=\sqrt{\frac{\sum_{i=1}^{n}{\left(y_{i,actual}-y_{i,predicted}\right)}^{2}}{n}}$$(2)$$R_{c}^{2},R_{p}^{2}=1-\frac{\sum_{i=1}^{n}{\left(y_{i,actual}-y_{i.predicted}\right)}^{2}}{\sum_{i=1}^{n}{\left(y_{i,actual}-{\bar{y}}_{actual}\right)}^{2}}$$(3)$$RPD=\frac{\sqrt{\frac{1}{n-1}\sum_{i=1}^{n}{\left(y_{i,actual}-{\bar{y}}_{actual}\right)}^{2}}}{RMSE}$$

In the formulas, $y_{i,actual}$ represents the reference measurement of the i-th sample, $y_{i.predicted}$ represents the predicted value of the i-th sample, ${\bar{y}}_{actual}$ denotes the mean reference measurement of all samples, and *n* is the number of samples in the dataset. RMSE is used to evaluate the prediction accuracy of the model, while R^2^ measures the model’s goodness of fit, typically ranging from 0 to 1. In addition, RPD is employed to assess the robustness of the predictive model. In general, an accurate and robust predictive model should exhibit a higher R^2^ and RPD, along with a lower RMSE.

### 2.9. Software

All algorithms were implemented on the Windows 11 operating system using the PyCharm 2023.2.1 (JetBrains, Prague, Czech Republic) environment with Python 3.11 (Python Software Foundation, Wilmington, DE, USA). The training and testing of deep learning models were conducted using the PyTorch 2.7.0 (Meta AI, Menlo Park, CA, USA) framework and executed on an NVIDIA GTX 1060 GPU (NVIDIA Corp., Santa Clara, CA, USA).

## 3. Results and Discussion

### 3.1. Distribution Characteristics of AFB_1_ Content and Sample Set Partitioning

In this study, a total of 330 maize samples with varying levels of AFB_1_ contamination were collected, with concentrations ranging from 1.153 μg·kg^−1^ to 76.521 μg·kg^−1^. These statistical measures are aligned with values reported in previous studies [46]. To expand the dataset size and diversity and enhance the generalization capability of the model, three data augmentation methods—multiplicative random scaling, bootstrap resampling, and WGAN—were employed to increase the dataset from 330 to 1320 samples. Subsequently, the SPXY algorithm was applied to partition the dataset into calibration and prediction subsets at a 7:3 ratio. Table 2 summarizes the AFB_1_ distribution characteristics and the corresponding dataset partitioning before and after augmentation.

### 3.2. Spectral Analysis

As shown in Figure 4, the raw spectral reflectance curves of all samples exhibit distinct differences in reflectance intensity, likely due to varying AFB_1_ concentrations among samples. The spectral curves display distinct peaks or troughs near 1153 nm, 1239 nm, 1349 nm, and 1477 nm. Among these, the peaks and troughs at 1153 nm and 1239 nm are associated with C-H groups [47]. The peak at 1349 nm corresponds to the combination stretching vibration of the C-H bond, while the trough at 1477 nm is related to the O-H group vibration and the structural characteristics of starch molecules, which are to some extent associated with the presence of *Aspergillus flavus* [19]. The observed peaks and troughs at these wavelengths are consistent with previous studies, further validating the reliability of the spectral detection results in this study.

### 3.3. Modeling Performance of AFB_1_ Prediction Under Combined Spectral Preprocessing and Data Augmentation

Due to the wide variability and uneven distribution of AFB_1_ concentrations in the original dataset, which reflects the inherent natural differences in aflatoxin contamination levels in maize, the RPD was lower than 1.4, indicating unreliable model calibration when using the raw spectra alone. Therefore, data augmentation was introduced to increase spectral diversity and strengthen the representativeness of the training samples. Importantly, all predictive evaluations in this study were performed against the reference AFB_1_ concentrations of real maize samples, ensuring reliable assessment of model accuracy.

After data augmentation and preprocessing, the predictive performance of the model under various preprocessing–augmentation combinations was systematically evaluated. As shown in Table 3, the prediction performance of the proposed MSFNet-ECA model under different combinations of spectral preprocessing methods, including none, D1, D2, SNV, MSC, and S-G smoothing, together with three data-augmentation techniques—multiplicative random scaling, bootstrap resampling, and WGAN—is summarized. The RPD values of the three data augmentation methods without preprocessing are 1.2, 1.2, and 1.3, respectively, all of which fail to meet the reliable modeling threshold of RPD > 1.4 [30]. This indicates that the constructed models exhibit weak predictive performance and limited generalizability. Therefore, employing data augmentation strategies alone under the current model is insufficient to achieve a significant improvement in predictive accuracy.

To further improve the model’s capability for predicting AFB_1_ levels in maize kernels and to overcome the performance plateau of single data augmentation strategies, this study incorporated several spectral preprocessing methods, including D1, D2, SNV, MSC, and S-G smoothing. These preprocessing techniques aim to suppress spectral noise, correct baseline drift, and improve spectral feature representation, highlight discriminative spectral features, thereby providing higher-quality input data for subsequent modeling. As shown in Table 3, the introduction of spectral preprocessing methods in conjunction with the three data augmentation approaches markedly enhanced the model’s predictive performance for AFB_1_ content in maize. Substantial variability in prediction accuracy was observed when different preprocessing methods were paired with various data augmentation strategies, emphasizing that effective preprocessing is crucial to maximizing the benefits of data augmentation.

In particular, among all preprocessing methods, the performance of multiplicative random scaling combined with D2 preprocessing achieved the best performance, with RMSEP of 2.3 μg·kg^−1^ and $R_{p}^{2}$ of 0.99. Compared with results without preprocessing, the model’s RMSEP decreased by 14.7 μg·kg^−1^, $R_{p}^{2}$ increased by 0.62, and RPD improved from 1.2 to 9, indicating that integrating preprocessing with data augmentation substantially enhanced the model’s predictive capability.

According to Table 3, the Bootstrap method shows strong modeling potential under different preprocessing conditions, particularly under D2 preprocessing, where the RMSEP is reduced to 6 μg·kg^−1^, $R_{p}^{2}$ increases to 0.92, and RPD rises to 3, demonstrating good quantitative performance. However, since the Bootstrap method essentially performs repeated sampling of the original data, it results in high inter-sample correlation. Moreover, when the total sample size is limited, the resampled data offer limited diversity in the feature space and may not effectively expand the sample distribution range. In addition, under its best-performing SNV preprocessing condition, WGAN still shows relatively weak modeling performance. Its RMSEP is 13 μg·kg^−1^, which is lower than that of the model constructed with unprocessed data; however, its $R_{p}^{2}$ is only 0.63, and RPD is only 1.6, indicating poor predictive ability. This implies that the representational quality and variability of the produced samples are not yet adequate for high-precision model construction.

As indicated by Table 3, several other commonly used spectral preprocessing methods exhibited varying degrees of impact on model performance. D1 showed consistently strong performance across all three data augmentation strategies, demonstrating its effectiveness in extracting spectral feature edges. SNV preprocessing also achieved relatively good results with multiplicative random scaling, indicating its significant role in correcting scattering interference. In contrast, MSC and SG generally exhibited moderate or suboptimal performance across most combinations and, in some cases, even weakened model performance, reflecting their weaker adaptability under different augmentation strategies.

Moreover, previous studies have demonstrated that data augmentation can effectively alleviate the strong dependence of deep learning models on sample quantity and distribution. For instance, Dianyang Sun employed WGAN to generate virtual wheat spectra and successfully enhanced the robustness of predictive performance by increasing sample diversity, while Syed Danish Ali confirmed that GAN-generated spectra can improve model generalization [31,32]. However, although WGAN has shown considerable potential in expanding spectral variability and mitigating distribution bias in earlier studies, the synthetic spectra in our experiment still exhibited distribution deviations and insufficient representation of fine-grained spectral structures, resulting in noticeable discrepancies compared with real maize spectra. Consequently, its contribution to AFB_1_ prediction performance improvement remained limited. This suggests that high-quality spectral generation may require more specialized generative network designs tailored to the complex scattering behavior and chemical absorption characteristics of maize kernels.

In addition to these performance differences, the present findings are also consistent with previous studies conducted on maize and other agricultural commodities. Yong-Kyoung Kim reported that scattering interference substantially affects the modeling accuracy in hyperspectral detection of aflatoxin contamination in maize, and that appropriate preprocessing is essential to ensure the effectiveness of modeling [19]. Likewise, Gayatri Mishra demonstrated that second-derivative processing can effectively enhance weak mold-related absorption features and improve prediction sensitivity in AFB_1_ quantification of almonds. In line with these results, the combination of D2 and multiplicative random scaling in this study achieved an outstanding RPD value of 9, confirming its capability to reduce scattering effects induced by the rigid structure of maize kernels and to strengthen spectral variations associated with aflatoxin content [20].

Overall, these results demonstrate a significant interaction between spectral preprocessing and data augmentation, highlighting the necessity of their collaborative optimization to maximize modeling capability and improve the accuracy of AFB_1_ prediction in highly heterogeneous maize samples.

### 3.4. Comparative Analysis of Modeling Performance Between Traditional Machine Learning Methods and Deep Learning Methods

To highlight the advantages of the MSFNet-ECA model in quantitative analysis, we adopted D2 preprocessing combined with multiplicative random scaling data augmentation. A comparative analysis was then conducted against four commonly used models: PLSR, SVR, ELM, and 1DCNN. The prediction results of each model are shown in Table 4, and their regression scatter plots are illustrated in Figure 5.

Table 4 shows that among PLSR, SVR, and ELM models, ELM performs relatively better, with RMSEP of 8 μg·kg^−1^, lower than that of PLSR at 17 μg·kg^−1^ and SVR at 11μg·kg^−1^. Moreover, the RPD of ELM is 2.4, higher than that of PLSR at 1.2 and SVR at 1.8. As shown in Figure 5a–c, ELM exhibits more outliers in the low-concentration region, showing its limitations in handling high-dimensional spectral features and nonlinear relationships. In contrast, the SVR model shows weaker performance, with predictions distributed sparsely across all concentration ranges, particularly with significant deviations in the high-concentration region. These characteristics limit its potential for high-precision quantitative analysis. Furthermore, PLSR shows an even weaker fitting ability in the high-concentration region, with scatter points significantly dispersed, indicating its major limitations in modeling complex nonlinear relationships and high-dimensional spectral features. In addition, 1DCNN—a commonly used deep learning model—demonstrates better performance, with RMSEP of 4 μg·kg^−1^, $R_{p}^{2}$ of 0.97, and RPD of 5, outperforming traditional machine learning methods. From the scatter plot in Figure 5d, the prediction results are generally distributed along the fitting line, but there are still some outliers in the high-concentration region, indicating that its fitting ability for high-concentration samples remains limited.

According to Table 4, the MSFNet-ECA model demonstrated the best performance. Compared with the PLSR, SVR, ELM, and 1DCNN models, the RMSEP of the MSFNet-ECA model decreased from 17 μg·kg^−1^, 11 μg·kg^−1^, 8 μg·kg^−1^, and 4 μg·kg^−1^ to 2.3 μg·kg^−1^, respectively. This corresponds to improvements in modeling accuracy by 86.4%, 79.1%, 71.3%, and 42.5%. The RPD also increased to 9, from 1.2, 1.8, 2.4, and 5, demonstrating the MSFNet-ECA model’s superior predictive and generalization performance. As shown in Figure 5e, it can be observed that the MSFNet-ECA model’s predicted values are concentrated near the ideal fitting line across the entire concentration range, with tightly distributed scatter points and almost no systematic deviation. This demonstrates excellent fitting accuracy and model stability. The enhanced performance of the MSFNet-ECA framework primarily arises from the integration of multi-scale convolution branches, which simultaneously capture local weak features and global long-range dependencies, enhancing spectral detail perception. Meanwhile, the ECA channel attention mechanism effectively strengthens key information extraction and reduces redundancy, improving the model’s focus and computational efficiency. In addition, adaptive feature fusion dynamically integrates multi-scale features through a weighted combination, thereby improving the model’s ability to recognize and represent key wavelength features. This mechanism further improves robustness and generalization performance.

To ensure that the observed improvement was not due to random variation, we further conducted a paired permutation test (100,000 sign-flips) on the absolute prediction errors over the SPXY test set. The MSFNet-ECA model achieved significantly smaller prediction errors than PLSR, SVR, ELM, and 1D-CNN, with average reductions of 12 μg·kg^−1^, 4 μg·kg^−1^, 4 μg·kg^−1^, and 1.2 μg·kg^−1^, respectively, and all *p*-values < 0.05. The distribution of prediction errors for each model is shown in Appendix A. These results confirm that the superior performance of MSFNet-ECA is statistically reliable and highly robust, rather than arising from random fluctuations or overfitting.

In recent research, Lina Guo integrated hyperspectral imaging with an SVR-SD-Mixup-PCA modeling framework for non-destructive quantification of AFB_1_ in maize silage, achieving a prediction coefficient $R_{p}^{2}$ of 0.95, RMSEP of 3 µg·kg^−1^ and RPD of 4 [7]. These results demonstrate that appropriate data augmentation and feature-reduction strategies can enhance the predictive capability of traditional machine-learning models to a certain extent. Another study revealed that transforming one-dimensional spectral sequences into two-dimensional MTF images and analyzing them using a 2D-MTF-CNN substantially strengthened spectral representation learning. Compared with its one-dimensional counterpart, this method achieved $R_{p}^{2}$ of 0.99, RPD of 15, and RMSEP of 1.4 µg·kg^−1^, indicating that introducing spatial structural information can further improve prediction performance [46]. Nevertheless, such approaches usually require additional spectral conversion steps and greater computational resources, which may limit their deployment in real-world applications.

Overall, the proposed MSFNet-ECA model demonstrated the highest predictive accuracy for estimating AFB_1_ concentrations in maize. By directly processing raw one-dimensional spectra and leveraging multi-scale convolutional branches together with an efficient channel-attention mechanism, the model can effectively capture deep and discriminative spectral features while maintaining a desirable balance between predictive performance and computational complexity. These advantages confirm the strong capability of MSFNet-ECA in modeling high-dimensional and nonlinear hyperspectral data, and highlight its practical potential for rapid and reliable monitoring of aflatoxin contamination in maize.

It is worth noting that traditional linear methods, such as PLSR, exhibited limited predictive performance in this study. This limitation primarily stems from the fact that the near-infrared spectral response of maize kernels is affected by complex light-matter interactions, such as scattering and path-length variations, which lead to nonlinear relationships between spectral intensity and chemical composition [39,48]. In addition, hyperspectral signals are jointly influenced by multiple factors, such as light scattering, surface texture, and the spatial heterogeneity of aflatoxin contamination, which further introduce nonlinear and coupled spectral variations [19,39]. Consequently, linear regression models, which assume that the relationship between spectral intensity and analyte concentration is additive and proportional, fail to capture these complex nonlinear interactions. In contrast, nonlinear and deep learning models such as the MSFNet-ECA proposed in this study can automatically extract hierarchical spectral representations and model high-order correlations among spectral bands, leading to substantially improved prediction accuracy and robustness. Therefore, the poor performance of linear models in this study further confirms that the relationship between hyperspectral features and AFB_1_ concentration is essentially nonlinear.

It is worth noting that this study was conducted using maize kernels of a single crop type, and the model’s performance across different varieties and growth environments has not yet been fully investigated. Although the experimental scope was limited to maize, the proposed MSFNet-ECA framework shows strong potential for extension to other aflatoxin-susceptible cereal grains and agricultural commodities. Future work will explore transfer learning or domain-adaptation strategies to address inter-species variations in spectral characteristics and sample heterogeneity, thereby enhancing the robustness and scalability of the model. Moreover, the full modeling workflow and network architecture proposed in this study can be directly adopted or fine-tuned by other researchers using their own hyperspectral datasets, enabling practical application of the model in different food safety monitoring scenarios.

## 4. Conclusions

This study verified that a deep learning model based on near-infrared hyperspectral data and enhanced by data augmentation can achieve accurate and efficient monitoring of AFB_1_ contamination in maize, and further demonstrated that the use of data augmentation strategies can significantly improve the predictive performance of hyperspectral quantitative models. In addition, a deep learning-based MSFNet-ECA model was proposed for accurate AFB_1_ prediction in maize. A systematic comparison of three data augmentation techniques—multiplicative random scaling, Bootstrap resampling, and WGAN—revealed that multiplicative random scaling combined with D2 preprocessing achieved the best performance in the MSFNet-ECA model, effectively enhancing the model’s prediction accuracy and stability. Furthermore, the MSFNet-ECA model constructed in this study achieved RMSEP of 2.3 μg·kg^−1^, $R_{p}^{2}$ of 0.99, and RPD value of 9 for AFB_1_ prediction. Compared with four commonly used quantitative analysis models—PLSR, SVR, ELM, and 1DCNN—the RMSEP of the proposed model was reduced by 86.4%, 79.1%, 71.3%, and 42.5%, respectively. In conclusion, this study demonstrates the effectiveness of collaborative optimization between data augmentation and preprocessing in deep learning modeling. It also highlights the superior ability of the proposed MSFNet-ECA model in handling high-dimensional nonlinear spectral data. The proposed approach delivers valuable insights and methodological guidance for applying NIR hyperspectral techniques to food quality and safety monitoring, indicating promising feasibility for practical implementation and extensive utilization.

## Figures and Tables

**Figure 1 foods-14-03769-f001:**
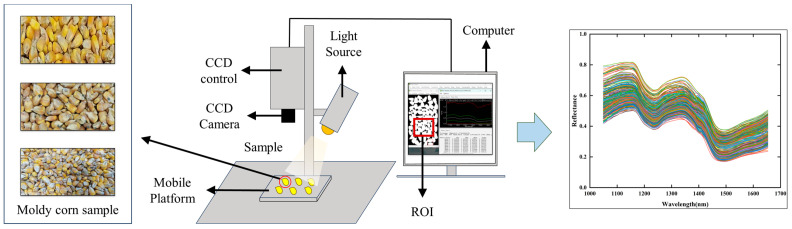
Schematic diagram of hyperspectral imaging and spectral extraction for moldy maize samples, illustrating the configuration of the imaging system, sample placement, region of interest (ROI) selection, and the obtained sample reflectance spectra.

**Figure 2 foods-14-03769-f002:**
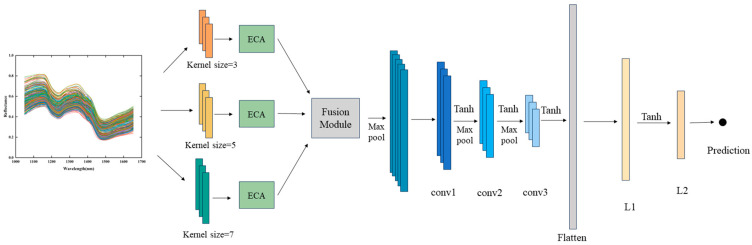
Schematic representation of the Multi-Scale Feature Network with Efficient Channel Attention (MSFNet-ECA) model architecture, illustrating three multi-scale convolution branches (kernel sizes of 3, 5, and 7) integrated with Efficient Channel Attention (ECA) modules, a feature fusion module, and regression layers designed for quantitative prediction of AFB_1_ content.

**Figure 3 foods-14-03769-f003:**
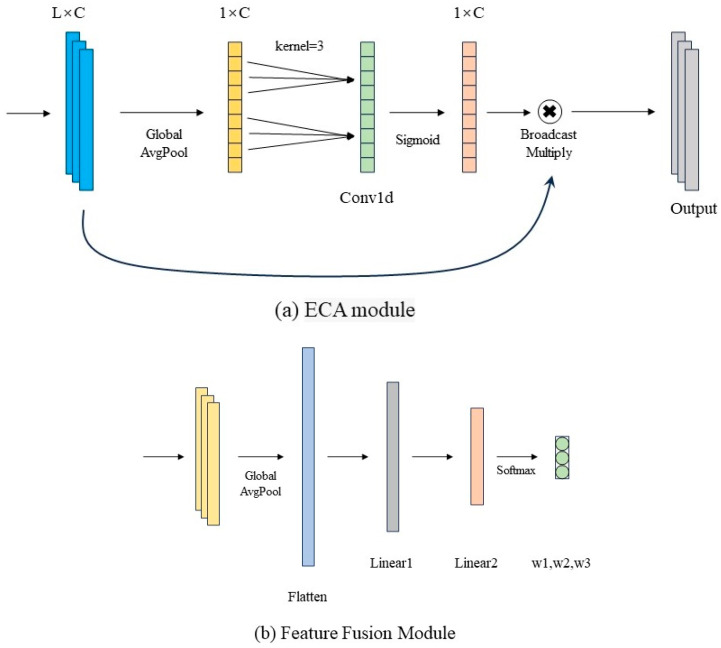
Structural illustration of key components in the MSFNet-ECA model: (**a**) the ECA module, which employs global average pooling and 1D convolution to capture inter-channel dependencies and generate adaptive attention weights; (**b**) the Feature Fusion Module, implemented as a multilayer perceptron that applies global pooling and Softmax-based weighting to adaptively integrate multi-scale spectral features.

**Figure 4 foods-14-03769-f004:**
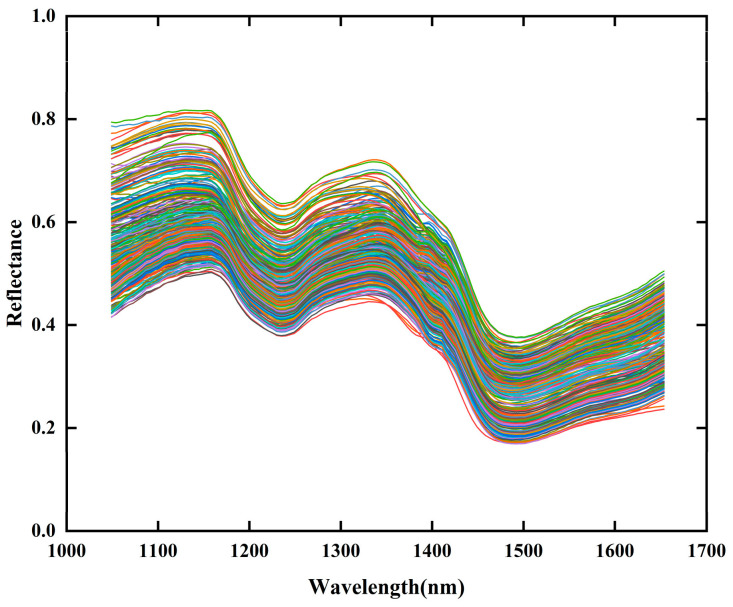
Near-infrared reflectance spectra of raw maize samples in the wavelength range of 1049.16–1653.68 nm, illustrating overall spectral variation among samples used for AFB_1_ prediction.

**Figure 5 foods-14-03769-f005:**
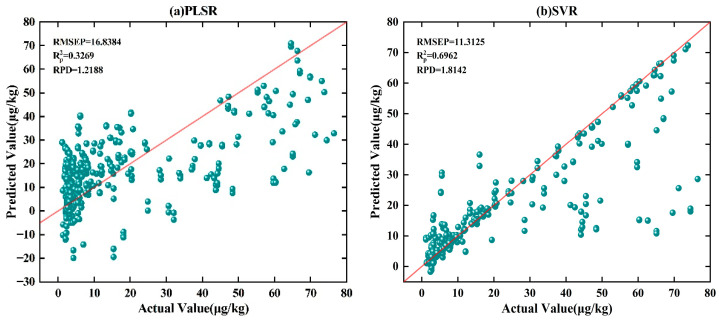
Regression scatter plots comparing the predicted and actual AFB_1_ concentrations in maize samples using different regression models. (**a**) partial least squares regression (PLSR), (**b**) support vector regression (SVR), (**c**) extreme learning machine (ELM), (**d**) one-dimensional convolutional neural network (1D-CNN), (**e**) MSFNet-ECA. The red line denotes the 1:1 reference line between the predicted and actual values.

**Table 1 foods-14-03769-t001:** Architecture configuration and key hyperparameter settings of the proposed MSFNet-ECA model for quantitative prediction of AFB_1_ content in maize.

Model	Layers		Size	Number
MSFNet-ECA	Multiscale	branch3	3 × 1	21
		branch5	5 × 1	21
		branch7	7 × 1	22
	ECA	Conv1d	3 × 1	1
	Feature Fusion model	Linear1	64	-
		Linear2	32	-
	Conv1		7 × 1	16
	Conv2		7 × 1	16
	Conv3		7 × 1	8
	L1		16	-
	L2		1	-

**Table 2 foods-14-03769-t002:** Statistical summary and SPXY-based partitioning results for the original dataset and the augmented datasets generated by different data augmentation methods.

Enhance Methods	Dataset	Samples	Max/μg·kg^−1^	Min/μg·kg^−1^	Mean/μg·kg^−1^	SD/μg·kg^−1^
Origin	Whole	330	76.521	1.153	17.44	20.60
Multiplicative	Whole	1320	76.521	1.153	17.44	20.57
	Calibration	924	76.521	1.153	17.49	20.60
	Prediction	396	76.521	1.153	17.32	20.55
Bootstrap	Whole	1320	76.521	1.153	17.29	20.52
	Calibration	924	76.521	1.153	17.26	20.53
	Prediction	396	74.527	1.153	17.35	20.52
WGAN	Whole	1320	76.521	1.153	17.44	20.57
	Calibration	924	76.521	1.153	17.47	20.57
	Prediction	396	76.521	1.153	17.37	20.60

**Table 3 foods-14-03769-t003:** Prediction performance of the proposed MSFNet-ECA model using different spectral preprocessing methods under three data augmentation techniques for AFB_1_ quantification in maize.

Methods	Pretreatment	RMSEC/μg·kg^−1^	$R_{c}^{2}$	RMSEP/μg·kg^−1^	$R_{p}^{2}$	RPD
Multiplicative	None	11	0.71	17	0.37	1.2
	D1	1.1	0.99	5	0.95	5
	D2	0.9	0.99	2.3	0.99	9
	SNV	3	0.98	3	0.97	6
	MSC	8	0.86	47	0.00	0.4
	SG	14	0.57	20	0.14	1.0
Bootstrap	None	16	0.38	18	0.26	1.2
	D1	5	0.93	9	0.79	2.2
	D2	2.1	0.99	6	0.92	3
	SNV	14	0.56	15	0.46	1.4
	MSC	13	0.62	20	0.17	1.0
	SG	16	0.38	18	0.26	1.2
WGAN	None	11	0.71	15	0.48	1.3
	D1	5	0.94	16	0.50	1.3
	D2	6	0.93	15	0.54	1.3
	SNV	6	0.90	13	0.63	1.6
	MSC	6	0.91	13	0.60	1.5
	SG	6	0.91	13	0.60	1.5

**Table 4 foods-14-03769-t004:** Comparative prediction performance of PLSR, SVR, ELM, 1D-CNN, and the proposed MSFNet-ECA model for quantitative analysis of AFB_1_ content in maize.

Model	RMSEC/μg·kg^−1^	$R_{c}^{2}$	RMSEP/μg·kg^−1^	$R_{p}^{2}$	RPD
PLSR	15	0.47	17	0.33	1.2
SVR	10	0.76	11	0.70	1.8
ELM	1.5	0.99	8	0.83	2.4
1DCNN	2.1	0.99	4	0.97	5
MSFNet-ECA	0.9	0.99	2.3	0.99	9

## Data Availability

The datasets collected and analyzed during the current study are not publicly available due to ongoing research and potential application restrictions, but are available from the corresponding author on reasonable request.

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
