# Peer review of "Multi-Scale Feature Attention Network for Rapid and Non-Destructive Quantification of Aflatoxin B1 in Maize Using Hyperspectral Imaging"

_foods, 2025, doi:10.3390/foods14213769_

Round 1
Reviewer 1 Report
Comments and Suggestions for Authors
The manuscript is solid and robust, with appropriate sections for the introduction, materials and methods, results, and discussion. However, the captions for all tables and figures require revision. They are overly simplistic, making it time-consuming to interpret the figures and tables. Captions should be rewritten to reflect all elements depicted and be fully self-explanatory. Table 1 contains only a single row, which is unusual and should be clarified. Please add statistical tests comparing the performance of the models. In the results section, clearly indicate which figure or table is being referenced. The combined results and discussion section needs a stronger theoretical framework to contextualise the findings. The authors should explicitly state whether the hypothesis was confirmed or not by the models used. Please include a paragraph comparing these results with data for the same species and, if applicable, discuss any differences observed with other cultivars. Is it possible to apply this method to other species? Would new models need to be developed, and if so, how should this be approached? Finally, clarify how other authors can apply the presented models in their own research.
Reviewer 2 Report
Comments and Suggestions for Authors
The manuscript entitled “Multi-Scale Feature Attention Network for Rapid and Non-Destructive Quantification of Aflatoxin B1 in Maize Using Hyperspectral Imaging” presents a highly relevant and technically sound study that combines hyperspectral imaging with deep learning for the accurate quantification of aflatoxin B1 in maize. The integration of the Multi-Scale Feature Network with Efficient Channel Attention represents an innovative approach to improving prediction accuracy and efficiency in non-destructive food safety monitoring. The topic is of great importance for agricultural and food security research, given the global concern over aflatoxin contamination and its health implications. Overall, the study is well-designed, methodologically rigorous, and clearly written, providing valuable insights into the application of advanced imaging and deep learning for rapid toxin detection. However, the following recommendations are suggested:
- In Section 2.1, it is recommended to include more detailed information about the experiments performed. Were maize seeds or kernels used? This is never mentioned in the manuscript. It is also unclear why the samples were placed on a metallic tray, how the evaluation was set up in the incubator (materials, conditions), where the mold was obtained from, how long each experimental cycle lasted, and whether each cycle involved a complete setup of the experiment from the beginning. What type of maize material was used, hybrid or variety? Why was only one type of material selected? Have the authors considered potential variations due to grain color differences, or color changes in the mold depending on the grain material used?
- It is recommended to improve all figure captions, as the information provided is insufficient and the figures are not self-explanatory.
- Section 2.3 seems to include results rather than methodology; why are these results presented as part of the methods section?
- In line 227, the description of Table 2 does not correspond to the data shown in Table 2; please revise this inconsistency.
- Section 2.6 contains many subsections that are not strictly methodological and should be moved to another section of the manuscript.
- The table headings are not informative and should be revised.
- In Table 3, please specify the units for “size.”
- The Results and Discussion section focuses mainly on results. It is recommended to reinforce the discussion by including more references and comparisons with previous studies.
- It is unclear how the analysis predicts AFB1 content, since there is no section comparing the model with the data obtained from the analysis of the 330 samples. Moreover, the data variation presented in Table 1 appears to be quite large.
- Please ensure that all scientific names are italicized (e.g., lines 40, 42, 377, etc.).
